# Connexin Gap Junction Channels and Hemichannels: Insights from High-Resolution Structures

**DOI:** 10.3390/biology13050298

**Published:** 2024-04-26

**Authors:** Maciej Jagielnicki, Iga Kucharska, Brad C. Bennett, Andrew L. Harris, Mark Yeager

**Affiliations:** 1The Phillip and Patricia Frost Institute for Chemistry and Molecular Science, Department of Chemistry, University of Miami, 1201 Memorial Drive, Miami, FL 33146, USA; mxj867@miami.edu (M.J.); ixk277@miami.edu (I.K.); 2Department of Biological and Environmental Sciences, Howard College of Arts and Sciences, Samford University, Birmingham, AL 35229, USA; bbennet1@samford.edu; 3Rutgers New Jersey Medical School, Department of Pharmacology, Physiology and Neuroscience, Newark, NJ 07103, USA; aharris@njms.rutgers.edu; 4The Phillip and Patricia Frost Institute for Chemistry and Molecular Science, Department of Biochemistry and Molecular Biology, University of Miami, Miami, FL 33146, USA

**Keywords:** connexin, gap junction channel, gap junction hemichannel, electron cryomicroscopy, X-ray crystallography, calcium regulation, pH regulation, channel gating, lipid binding

## Abstract

**Simple Summary:**

Gap junction channels are composed of an assembly of connexin proteins and allow direct communication between cells. They are highly conserved across vertebrates and form wide pores in cell membranes for the passage of ions and metabolites. Junctional channels are formed from the end-to-end docking of hemichannels, and both junctional channels and hemichannels are vital for many physiological activities. Several medical conditions are associated with problems in gap junction communication, ranging from deafness to fatal cardiac arrhythmias. Many connexin channel diseases can be linked to genetic mutations, and nearly 1000 have been identified in connexin genes. Prior to 2009, atomic-level structural details of gap junction channels were essentially non-existent. This information is critical for understanding channel function and to assess the pathological nature of disease-causing mutations. Fortunately, since 2009, the powerful tools of X-ray crystallography and electron cryomicroscopy have yielded over 50 high-resolution structures of connexin channels. This review aims to provide a comprehensive summary of this astounding 15-year period of structural discovery in the gap junction field. Divided into eight distinct sections, we describe key details found in this compendium of structures, such as conserved features in the design of connexin channels, insights into channel gating and surprises regarding where membrane lipids are bound to the channels. In addition, we highlight areas in which we need more information, such as the structure of highly flexible regions within connexin channels that have so far resisted visualization. Furthermore, targeting connexins for drug discovery is still in its infancy, and much more structural data are needed to pursue this end.

**Abstract:**

Connexins (Cxs) are a family of integral membrane proteins, which function as both hexameric hemichannels (HCs) and dodecameric gap junction channels (GJCs), behaving as conduits for the electrical and molecular communication between cells and between cells and the extracellular environment, respectively. Their proper functioning is crucial for many processes, including development, physiology, and response to disease and trauma. Abnormal GJC and HC communication can lead to numerous pathological states including inflammation, skin diseases, deafness, nervous system disorders, and cardiac arrhythmias. Over the last 15 years, high-resolution X-ray and electron cryomicroscopy (cryoEM) structures for seven Cx isoforms have revealed conservation in the four-helix transmembrane (TM) bundle of each subunit; an αβ fold in the disulfide-bonded extracellular loops and inter-subunit hydrogen bonding across the extracellular gap that mediates end-to-end docking to form a tight seal between hexamers in the GJC. Tissue injury is associated with cellular Ca^2+^ overload. Surprisingly, the binding of 12 Ca^2+^ ions in the Cx26 GJC results in a novel electrostatic gating mechanism that blocks cation permeation. In contrast, acidic pH during tissue injury elicits association of the N-terminal (NT) domains that sterically blocks the pore in a “ball-and-chain” fashion. The NT domains under physiologic conditions display multiple conformational states, stabilized by protein–protein and protein–lipid interactions, which may relate to gating mechanisms. The cryoEM maps also revealed putative lipid densities within the pore, intercalated among transmembrane α-helices and between protomers, the functions of which are unknown. For the future, time-resolved cryoEM of isolated Cx channels as well as cryotomography of GJCs and HCs in cells and tissues will yield a deeper insight into the mechanisms for channel regulation. The cytoplasmic loop (CL) and C-terminal (CT) domains are divergent in sequence and length, are likely involved in channel regulation, but are not visualized in the high-resolution X-ray and cryoEM maps presumably due to conformational flexibility. We expect that the integrated use of synergistic physicochemical, spectroscopic, biophysical, and computational methods will reveal conformational dynamics relevant to functional states. We anticipate that such a wealth of results under different pathologic conditions will accelerate drug discovery related to Cx channel modulation.

## 1. Introduction

Connexins (Cxs) are a family of integral membrane proteins that assemble as hexameric hemichannels (HCs), which can dock end-to-end between apposing cells to form dodecameric, gap junction channels (GJCs) with a ~15 Å intercellular pore that enables the exchange of metabolites and signaling molecules up to roughly 1 kDa (Figure 1A,B). These channels are physical conduits that mediate direct electrical and chemical communication between adjacent cells [1]. At the junctional interface of the apposed cells, GJCs can cluster to form dense, quasi-hexagonal arrays called gap junction plaques (Figure 1C) [2]. A wide variety of ions and metabolites, from K^+^ to ATP, can transit through GJC intercellular pores [3,4].

The various Cx isoforms can combine into both homomeric and heteromeric gap junctions (Figure 2), which may exhibit different functional properties, including pore conductance, size and charge selectivity, as well as voltage and chemical gating [6]. Cx channels are the most well-studied of the four-helix bundle wide-pore transmembrane channels, a family of structurally homologous membrane proteins expressed in vertebrates [7]. Cx homologs also exist in invertebrates (the innexins). A unicellular form in protozoa (the unnexins) has recently been reported [8].

Cx monomers adopt a tertiary structure composed of four α-helical transmembrane (TM) domains (M1–M4), two extracellular loops (E1 and E2), a cytoplasmic loop (CL) connecting M2 to M3, and cytoplasmic N- and C- terminal domains (NT and CT) (Figure 3A,B) [10]. The docking of HCs to form GJCs is mediated by the E1 and E2 loops (the ECD), forming an extracellular vestibule bounded by a wall of protein, which forms a tight seal that excludes extracellular ions and small molecules (Figure 1A, 4). The direct intercellular exchange of hydrated ions, second messengers, and other cytosolic molecules is critical for the coordination of cellular events, from differentiation to synchronicity. For example, Cx36 GJCs are required for proper neuronal synaptic transmission [11] while Cx40 and Cx43 GJCs are necessary for ion conduction between cardiomyocytes in the heart, which regulates the heartbeat to coordinate ventricular contraction. Point mutations in Cx40 and Cx43 genes can cause aberrant conduction through GJCs and lead to potentially life-threatening atrial and ventricular arrhythmias [12]. 

As there is a diversity of Cx isoforms with specialized functions, HCs and GJCs of different compositions possess different permeability profiles, with charge preferences and restrictions on the size and shape of cargo that may pass. A general rule for molecular permeability of GJCs is that the pore MW cut-off is ~0.6 kDa with a size limit of 12–15 Å [4]. The tight regulation of GJC and HC function is necessary, as aberrant opening/closing is associated with pathological states and can compromise cell viability. Regulatory factors can have significant effects on channel activity including voltage, pH, phosphorylation, membrane lipid composition, and divalent cation binding [3,14,15]. Over 930 mutations or variants have been identified thus far amongst 11 Cx isoforms, and more mutations are certain to be found in other isoforms. Many of these Cx gene mutations are linked to over two dozen disparate diseases and dysfunctional states, from deafness, skin diseases, and cataracts to atrial fibrillation and arrhythmias, reflecting the widespread and specific distributions of Cx isoforms in different tissues [16,17]. For example, mutations of the *GJB1* (Cx32) gene on the X chromosome can cause Charcot–Marie–Tooth X1 (CMTX1) disease, typified by demyelination and axon loss in neurons in the peripheral nervous system [18]. Over 150 point mutations have been identified in Cx26 alone, each resulting in syndromic or non-syndromic deafness [17,19]. Mutations in Cx26 are the single most common cause of non-syndromic deafness. Our understanding of connexin channel function in physiological and pathophysiological contexts is rich. However, structures at near-atomic resolution are needed to allow the understanding of the mechanistic details that will inform the functional consequences of regulatory factors and point mutations. The ability to exploit connexins as potential therapeutic targets underscores the importance of obtaining high-quality structural data to facilitate drug design.

The expression of monomers, folding and oligomerization in the sarcoplasmic reticulum, transport to the plasma membrane, assembly of gap junction channels and association into plaques are a separate focus of research in the field, which has been thoroughly reviewed [20,21]. It is surprising that the tenacious attachment of hemichannels results in turn over as double-membrane vesicles containing intact GJCs [22]. This requires that part of the plasma membrane in the apposed cell is moved into the companion cell. Remarkably, gap junction channels turn over with a half-life ranging from 1–5 h [23,24,25].The process of assembly and turnover must be performed with incredible fidelity; otherwise, we would all succumb to sudden death from fatal cardiac arrhythmias!

Because GJC plaques form dense quasi-hexagonal arrays in the plasma membrane, it was possible in the mid-1980s to obtain low-resolution three-dimensional image reconstructions of GJCs via 2D electron crystallography, which proved the dodecameric architecture of GJCs [26]. In 1999, electron cryomicroscopy (cryoEM) of two-dimensional crystals of GJCs assembled from recombinant, human Cx43 yielded the first subnanometer resolution maps (first at 7.5 Å and then 5.7 Å in-plane resolution), revealing that each hemichannel is comprised of 24 TM α-helices [27,28]. In 2007, electron crystallography of the recombinant Cx26 M34A deafness mutant confirmed the modeling of the Cx43 TM helices and also identified a “plug” of density at the cytoplasmic vestibule attributed to an oligomer of the amino tails (NTs) [29].

Nevertheless, no experimentally determined, atomic resolution models of GJC channels existed until 2009, when the Cx26 GJC structure was solved by X-ray crystallography at 3.5 Å resolution [13,30]. The X-ray structure recapitulated the architecture of the Cx43 GJC derived from native sources [27,28], revealed the molecular details of TM domain organization, and mapped the network of hydrogen-bond interactions in the extracellular loops E1 and E2 responsible for the docking of hexameric hemichannels (Figure 4). Subsequent computational studies using the Cx26 model (MD equilibration followed by Grand Canonical Monte Carlo Brownian Dynamics simulations) reproduced experimentally determined current–voltage relations following the incorporation of MS-based co- and posttranslational modifications [31]. In 2016, two X-ray structures of the Cx26 GJC with improved statistics were solved with and without bound Ca^2+^, at 3.3 and 3.8 Å resolution, respectively [13,30]. The structures revealed that 12 Ca^2+^ ions are coordinated between adjacent subunits in each HC at the TMD/ECD boundary. The Cα positions in the two structures were nearly identical, ruling out a steric model for Ca^2+^ gating; i.e., previous low resolution cryoEM maps of liver gap junction plaques suggested that Ca^2+^ gates GJCs via the sliding and tilting of adjacent subunits, analogous to a camera iris [26,32]. Instead, analysis of the high-resolution X-ray structures suggested that Ca^2+^ binding creates an electrostatic barrier to permeation by cations [13,30]. In 2018, the first cryoEM structure of heteromeric Cx46/50 GJCs derived from native sources (sheep lens) was determined using single-particle image analysis. The map at ~3.5 Å rivaled those of the earlier X-ray crystal structures [33]. Thereafter, all subsequent high-resolution GJC structures have been solved by the single-particle analysis of cryoEM images.

At the time of this review, a total of 58 cryoEM maps and 51 models have been deposited in the Electron Microscopy Data Bank (EMDB) and the Protein Data Bank (PDB), respectively, all since 2009 (Table 1). Nearly all of the deposited structures are at resolutions better than 4 Å, with the highest resolution maps at 1.9 Å. The structures of seven Cx isoforms, in both or either of their GJC and HC oligomeric states, have been solved, representing the Cx groups α (Cx43, Cx46/50), β (Cx26, Cx32), γ (Cx31.3), and δ (Cx36).

CryoEM has allowed structural analysis in a variety of environments that mimic native lipid bilayers, such as GJCs isolated from native sources, solubilized in detergents and polymers (e.g., amphipols), and reconstituted in lipid bilayer nanodiscs. The single-particle image analysis of electron micrographs over a range of symmetries and with focused classification and refinement revealed conformational heterogeneity. For example, the crystallographic and microscopic analyses captured distinct conformations of the NT domain, resulting in structures in a variety of open, “closed” (better described as occluded), and partially closed states, described in detail below. In this review, the GJC structures are classified into these three groups based on both the positioning of the NT domain and the limiting pore diameter: open structures have no NTs resolved or NTs lining the channel pore, with pore diameter of over ~13 Å; partially closed structures have NTs present, restricting pore size to under ~13 Å; and “closed” (occluded) structures have full blockage of the channel pore regardless of the presence of NTs (Table 1). Throughout this review, the limiting (narrowest) width was determined from the PDB entries as that between identical atoms of opposite subunits in the narrowest section of the pore (as seen in [13]). These may differ from the distances in the papers reporting the structures.

The use of diverse experimental and computational approaches has allowed acquisition of structural information to provide mechanistic insight into gating by Ca^2+^ [13,36] and pH [14] as well as structural changes in response to pCO_2_ [34]. Where the NTs have been resolved in the 6-fold symmetrized cryoEM maps and crystal structures, they adopt different conformations within the pore itself, highlighting its potential role in channel regulation. In some structures the NTs are unresolved, presumably due to intrinsic flexibility. Of particular interest, the cryoEM structures have captured elongated densities ascribed to lipids, between adjacent subunits and unexpectedly inside the Cx channel pore and intercalated between TM α-helices. Although the resolution of these lipid-like densities often prevent molecular modeling of bound lipids and/or detergents, the presence of these features provides intriguing possibilities regarding gating [38], the stabilization of NT conformations [36,38,39], and the interactions of Cx proteins with the surrounding membrane environment [41] that would be otherwise unavailable.

High-resolution structures of seven Cx isoforms mark the emergence of a new age of gap junction channel research. This review spans 2009–2024 and summarizes the major discoveries from this impressive 15-year period of structural productivity and is organized into eight sections: (1) conservation of the TMD and ECD architectures across seven Cx isoforms, (2) sequence diversity and disorder in the CL and CT domains, (3) gating in high Ca^2+^ mediated by an electrostatic mechanism, (4) gating at acidic pH mediated by a steric “ball-and-chain” mechanism, (5) the NT visualized in the channel pore adopts multiple conformations, (6) visualization of lipid-like densities in unexpected places, within the pore and/or the TMD helices, (7) functional implications for gating and permeation, and (8) a summary of structure/function correlations and insights from high-resolution structures.

## 2. Major Insights from Recent High Resolution Structures of Cx Channels and Hemichannels

### 2.1. Conservation of the TMD and ECD Architectures across Seven Cx Isoforms

Using the Cx26 model as a reference, the four α-helix bundle topology of the TMD (Figure 1A and Figure 3A,B) is closely conserved in all structures so far determined (Figure 5A). The M1 and M2 α-helices line the aqueous pore and are thus partially responsible for the molecular size restriction at the channel entrance (Figure 1B). M1 displays the highest conformational variability (Figure 5A), despite it having relatively high sequence identity among different Cx isoforms (Figure 5B).

The E1 and E2 extracellular loops display a mixed α-helical and β-sheet topology (Figure 3A,B). Interestingly, the E1 loops display high sequence identity and a near identical topology in X-ray and cryoEM maps (Figure 5A,B). We infer that these conserved features are due to the crucial role of the extracellular loops in docking two hexamers to create the GJC and in forming a tight molecular seal that precludes accessibility to the extracellular environment. The cryoEM maps of the Cx channels usually display the highest (best) local resolution in the ECDs followed by the TMDs. The NTDs have comparatively low resolution, and the CL and CT domains are not visualized.

The ECD (comprised of loops E1 and E2) is a vital component of GJCs. Situated on the extracellular-facing side of the plasma membrane, these loops are crucial for the engagement of apposed HCs, and thus the formation and regulation of GJCs (Figure 4). Both loops extend about 15 Å, spanning the extracellular gap between cells, from which the GJC name is derived. Sequence alignment of the E1 and E2 loops of all connexins known to make GJCs revealed that they are highly conserved [42]. The positions of E1 and E2 are usually stabilized by three inter-loop disulfide bonds. Indeed, in E1, most Cx isoforms exhibit a pattern of Cys-X_6_-Cys-X_3_-Cys, while in E2, they display Cys-X_4_-Cys-X_5_-Cys (Figure 3A). Notably, the sole exception is Cx31.3, which has Cys-X_5_-Cys-X_5_-Cys spacing [43,44] and has only been found to form functional HCs and not GJCs [6].

Functional and mutational studies have provided exceedingly valuable insight into the gating and regulatory properties of various GJCs. These studies have also explored the docking of HCs at their extracellular interfaces to form GJCs of homotypic or heterotypic assembly (i.e., HCs of the same or different compositions, respectively). The atomic details of HC–HC docking were first revealed in the X-ray structures of homotypic Cx26 GJCs [13,30]. To form the junctional channel, E2 loops of opposing Cx subunits form hydrogen bonds, whereas an E1 loop from one Cx subunit participates in a hydrogen bonding network with two E1 loops from the opposing subunit [13,30] (Figure 4). This docking mechanism is consistent amongst all Cx structures so far reported with adequate resolution, with variations in the residues involved in hydrogen bonding at the HC interface, including Cx43 [39], Cx36 [38], and Cx46 [41]. Interestingly, even a single mutation of selected residues forming hydrogen bonds at the HC interface can prevent the formation of GJCs in at least two isoforms (Cx26 and Cx32) [35,45].

**Figure 5 biology-13-00298-f005:**
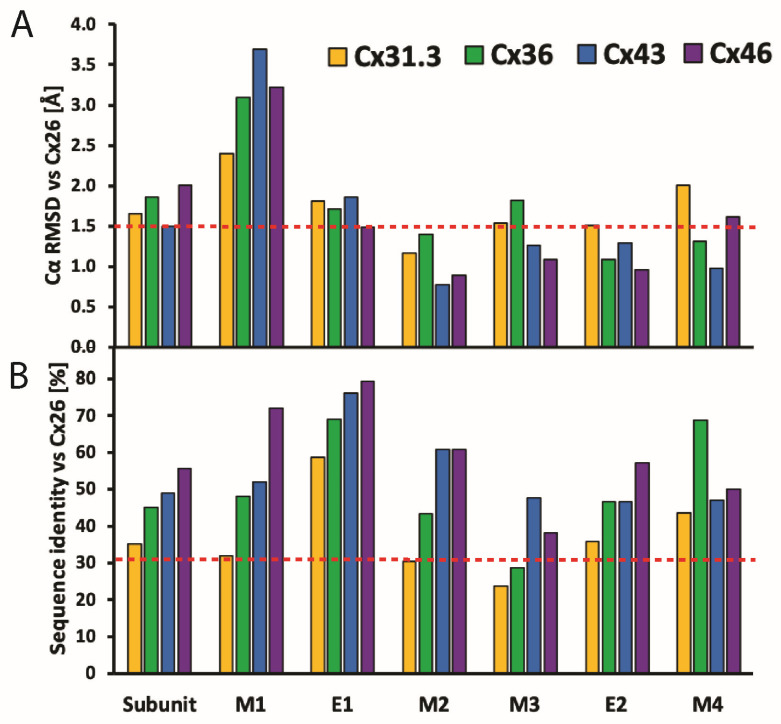
**Comparison of structural and sequence homology of Cx isoforms.** (**A**) Cα RMSD-based Cx subunit comparison, with Cx26 [13] (PDB: 5ER7) as a reference structure. The analyses include Cx31.3 [36] (PDB: 6L3T), Cx36 [38] (PDB: 7XNH), Cx43 [39] (PDB: 7F92), and Cx46 [41] (PDB: 7JKC). Dashed orange line denotes Cα RMSD value of 1.5 Å, below which structures are considered identical. Protein alignment performed on Cα atoms in PyMOL [46] using command “super” with parameter “cycles = 0”; RMSD calculation performed on Cα atom pairs in UCSF Chimera [47] using “rmsd” command. (**B**) Comparison of Cx subunits based on sequence identity, with Cx26 [13] (PDB: 5ER7) as a reference structure. Dashed orange line corresponds to sequence identity of 30%, below which subunits are considered significantly different in amino acid sequence. Analysis was performed with Clustal Omega [48] and includes Cxs sequences listed in (**B**).

### 2.2. Sequence Diversity and Disorder in the CL and CT Domains

Both the M2–M3 CL loop and the CT domain are unresolved in available structures, presumably due to their intrinsic disorder. The CT domain has been studied more and is a primary locus for biochemical regulation and intermolecular interactions of Cx channels [49,50,51,52,53]. Unlike the highly conserved TM and NT domains, there are major differences in the length and sequence of the CT domains among different connexins. The length of the CT ranges from 18 residues in Cx26 to 275 residues in Cx57 [54]. The CT of Cx43 (150 residues) is by far the best characterized [55]. The prediction of Cx43 CT secondary structure by nuclear magnetic resonance (NMR) spectroscopy [56] suggested the presence of specific α-helical and β-sheet regions. Nevertheless, ~70% of the CT is disordered, containing multiple linear motifs that allow for the potential interaction with a variety of proteins [55]. The CT also has a role in channel conductance.

The unitary conductance among Cx channels ranges from 15 to 300 pS, seemingly independent of ion selectivity [57]. Despite the availability of numerous GJC structures, the structural determinants responsible for this wide-ranging channel conductance remain unclear. The existing conserved architecture seems insufficient to explain this phenomenon. It is speculated that the structurally disordered CT domain might affect gating. In the case of the wide-pore channel pannexin (Panx1), a structurally disordered C-terminal tail penetrates the pore and obstructs the central entrance, limiting channel conductance. Caspase cleavage or genetic truncation of the Panx1 C-terminal tail dramatically enhances channel conductance, permitting the passage of large molecules such as ATP [58]. Similarly, the CT domain of Cx36 is involved in channel gating and regulation in response to the intracellular environment [59]. Although no structural data exist, the CT domain (in Cxs with long CT domains such as Cx43 or Cx40) functions as a gating particle, with full or partial Cx pore closure in response to various stimuli [60,61,62].

### 2.3. Gating in High Ca^2+^ Mediated by an Electrostatic Mechanism

Cytoplasmic Ca^2+^ was perhaps the first known regulator of junctional coupling [63], but the junctions in which this was shown were formed by members of the innexin family of proteins, distinct from connexins, which form gap junctions in invertebrates [64]. However, similar regulation by the cytoplasmic Ca^2+^ of GJCs formed by connexins was shown later [65,66,67], occurring over a 0.1–1.0 mM range. This work inspired the early EM crystallographic studies of native GJC plaques in the presence and absence of Ca^2+^, where Ca^2+^ was thought to induce an iris-like narrowing of the channel pore [26,32].

In other studies, it was shown that the open probability of undocked connexin HCs in plasma membranes was near zero at physiological concentrations of the extracellular Ca^2+^ (1.8–2.0 mM) and increased with reductions below this level, typically reaching a maximum near 0.1 mM [68,69,70] (reviewed in [71]). The Ca^2+^ dependence of undocked HCs has been well characterized in functional studies, and the site of Ca^2+^ binding that results in closed HCs has been identified [72,73,74].

The regulation of Cxs by Ca^2+^ has been investigated in the structures of Cx26 GJCs [13] and Cx31.3 HCs [36]. In the case of Cx26, two crystal structures of a GJC were obtained—one with and one without the addition of Ca^2+^. Contrary to what was observed previously in the low-resolution studies of GJCs [26] and HCs [75], the binding of Ca^2+^ did not cause large-scale conformational changes such as channel pore constriction by a “camera iris” mechanism. Instead, there were local changes around the inter-subunit Ca^2+^ binding sites at the boundary of M1 and E1, where Ca^2+^ was coordinated by residues G45 and E47 of one subunit and E42 of the adjacent subunit. This site of inter-subunit Ca^2+^ binding overlaps with that identified from the physiological and computational studies of Cx26 and Cx46 HCs [72,73]. Ca^2+^ binding utilizes a pentamer of protein ligands, with E47 and E42 contributing four coordinating carboxylate oxygens and G45 providing one carbonyl oxygen. In total, 12 Ca^2+^ ions were detected binding in a GJC, causing a dramatic change in channel pore electrostatics by creating positive surface potential (Figure 6A). MD simulations revealed a block of K^+^ permeation as a result of this Ca^2+^ binding [13,30]. The binding of Ca^2+^ was thus proposed to act as an electrostatic switch that dramatically shifts the charge selectivity of the channel, creating a positive potential inside channel pore. The region around the identified Ca^2+^ binding site, consisting of residues 40–50, is highly conserved among Cxs. Therefore, the electrostatic switch is proposed as a general mechanism for Ca^2+^ regulation in GJCs, with a Ca^2+^ binding motif of E41/42(subunit B)-G45(subunit A)-E/D47(subunit A) [13]. 

The cryoEM structures of Cx31.3 HC with and without Ca^2+^ also failed to show any major conformational changes [36]. The structure without Ca^2+^ identified multiple water molecules, some of which were bound at the M1/E1 boundary, close to extracellular end of the channel pore. The pore has a negative surface potential in that region, a feature conserved among Cxs. After the addition of Ca^2+^, no major structural rearrangements were observed, with apparently no narrowing of the pore. No bound ions could be detected unambiguously anywhere in the HC structure. However, some densities previously attributed to six water molecules bound at that M1/E1 boundary location became smeared in the +Ca^2+^ cryoEM map. Taken together with observed small amino acid side chain rearrangements in that region, putative Ca^2+^ binding sites were proposed that are more extensive but do overlap with those identified in the Cx26 GJC X-ray structure [13]. In Cx31.3, an inter-subunit tunnel hosting water and, possibly, Ca^2+^ binding sites, is formed by 17 conserved residues that are hot-spots for disease mutations [36]. The proposed location of Ca^2+^ binding in the tunnel is very close to the location where Ca^2+^ binding was shown in the Cx26 GJC X-ray structure [13]. The residue that underwent the most change upon addition of Ca^2+^ was E47; its density increased, and its side chain conformation shifted. This structural change might be due to the binding of Ca^2+^ followed by repulsion by two adjacent residues, R75 (same subunit) and R184 (neighboring subunit). Since amino acid changes in Cx31.3 HC upon Ca^2+^ binding are small, an electrostatic mechanism for Ca^2+^ regulation was proposed [36], mimicking that of Cx26 [13].

Strikingly, the X-ray [13] and cryoEM [36] studies identify sites of Ca^2+^ interaction within GJs (Cx26 and Cx31.3, respectively) that are consistent with each other, and with the region at which Ca^2+^ binds in HCs. Thus, there is a correspondence between the site of Ca^2+^ interaction determined by functional studies and the site of Ca^2+^ interaction seen in the X-ray and cryo-EM structural analyses of GJCs. One must be cautious that even if the site of Ca^2+^ interaction is essentially the same in HCs and GJCs, the channel is in different structural states and the consequences of Ca^2+^ binding may differ. At present, the electrostatic mechanism for the gating of GJCs in high Ca^2+^ is an appealing hypothesis [13] that awaits additional validation.

### 2.4. Gating at Acidic pH Mediated by a Steric “Ball-and-Chain” Mechanism

Cytoplasmic acidification was one of the first identified regulators of connexin channels [76,77]; the lowering of cytosolic pH (e.g., by one pH unit) by any number of means rapidly closes the channels, and the recovery of cytosolic pH permits the recovery of channel activity. The regulatory mechanism, which is nearly universal across all the connexin isoforms, and which is of likely biomedical importance, has remained largely opaque until recently. The studies of [14] show that under modestly acidic pH, the pore of Cx26 channels is occluded by protein density composed of the NT domains. In contrast, in neutral pH conditions, an open conformation is observed with reduced NT density resulting in an unobstructed pore (Figure 6B). The open conformation structures have residual densities in the channel pore, attributed to the α-helical portion of the NT domain [14], in a position similar to that reported in other Cx26 GJC structures [30,34]. In the occluded conformation, there are six threads of density emanating from the cytoplasmic ends of the M1 helices that extend into the channel pore, culminating in a plug within each HC. The volume of the plug with the connecting density threads matches that of six NTs, and thus, the occluding density is interpreted as fully extended NTs coming together to obstruct the pore (Figure 6B). Mass spectrometry (MS) has shown that NTs are acetylated at the Met1 residue [78], which neutralizes the positive charge of the N-terminal amino group and likely facilitates the association of the NTs. This study does not suggest a specific mechanism by which NTs separate from the walls of the pore and form a globular density that occludes the pore. The NT domain transitions resulted from the single perturbation of lowering the pH from 7.5 to 6.4. In addition, only the open state was observed at pH 7.5, whereas at pH 6.4 there was a distribution between the open and occluded states suggesting a two-state process. These results provide strong evidence that this is a physiological gating transition.

As background, in 1977, Armstrong and Bezanilla [79] showed that inactivation gating of Na^+^ channels was sensitive to proteolytic digestion. Then, in 1990, Aldrich and colleagues used a molecular biology approach to identify residues in the Shaker K^+^ channel that conferred inactivation [80]. Lastly, in 2001, Mackinnon and colleagues provided physiologic results and a composite model including the structure of the KcsA K^+^ channel to delineate the “ball-and-chain” mechanism for inactivation [81]. By analogy, Khan et al. proposed a similar steric “ball-and-chain” mechanism for gating the Cx26 GJC at acidic pH (Figure 6B) [14]. At physiologic pH, the NTs adopt multiple conformations with a more ordered α-helical region. At acidic pH, extension and association of the NTs generate a globular gating particle that occludes the pore. The results were confirmed by proximity relationships derived by MS. Hydrogen/deuterium exchange-MS (HDX-MS) showed that the amide hydrogens within the NTs did not exchange with D atoms as occurred at physiologic pH. This implied that the NTs were sequestered by association in forming the globular blocking particle. Likewise, crosslinking-MS (XL-MS) showed no crosslinks between the NTs and the M2–M3 cytoplasmic loops at physiologic pH, whereas several crosslinks were detected at acidic pH.

The occluding density shown in Figure 6B is seated at a similar depth in the channel pore as the plug reported in a cryoEM study of 2D crystals of Cx26 [29], which was not connected to any of the pore-lining helices. This plug density was speculated to be composed of the NT domain(s) based on the studies of an NT-deletion mutant of Cx26, which displayed reduced plug density [82]. It is notable that these previous studies [82] were conducted in conditions favoring closed channel conformation (mutant protein, low pH, Ca^2+^/Mg^2+^, carbenoxolone), whereas the more recent study examined the single variable of acidic pH [14].

### 2.5. The NT Visualized in the Channel Pore Adopts Multiple Conformations

On the basis of sequence homology, the NT domain is the most conserved cytoplasmic domain in Cxs, and its importance is emphasized by the fact that mutations in this domain are linked to a number of diseases including sensorineural deafness, skin diseases, neuropathies, and cataracts [17,19,83,84,85]. The highest number of disease mutations are in some of the most well-conserved positions, namely positions 12 (in β Cx isoforms)/13 (α Cxs) and 22 (β Cxs)/23 (α Cxs) (Table 2). Mutations can result in the dysregulation of Cx trafficking to the plasma membrane, defects in channel properties, and even the complete loss of function [83]. Cx channels are regulated by transjunctional voltage (Vj), which, depending on the regulatory mechanism, is referred to as either Vj-gating (fast-gating) or loop-gating (slow-gating). The NT appears to be the key mechanistic component of Vj-gating and determines its gating polarity, which can be either positive or negative, depending on the Cx isoform [86]. In β Cxs, the gating polarity and voltage sensitivity is determined by the charge of the first 10 amino acid residues, and the gating polarity can be reversed by single amino acid mutations at specific positions within this section of the NT. For example, positions 2, 5, and 8 are crucial for determining gating polarity [15,87,88,89]. Similarly, α Cxs also regulate their gating polarity and voltage sensitivity based on the charge of residues in the NT. As in β Cxs, the gating polarity can be reversed, or sensitivity to voltage altered, by single amino acid mutations at selected positions within the NT. Positions 3 and 9 (equivalent to 2 and 8 in β Cxs) are major determinants of the response to voltage [90,91,92]. Interestingly, these NT residues associated with voltage sensitivity are not well conserved, with the exception of Asp at the 2/3 position, which is present in about half of the β Cxs and in all α Cxs (Table 2). The lack of conservation at most of these key positions linked to NT-mediated voltage gating is likely responsible for conferring unique properties to channels formed by different Cx isoforms [83]. In addition to its importance in Vj-gating, numerous studies of point mutations and chimeric constructs, supplemented by computational studies, make clear that the NT is a determinant of multiple aspects of Cx channel physiology (e.g., open unitary conductance, stability of open and subconductance states, charge selectivity, subunit interactions, and overall dynamics [93,94,95,96,97]).

Even though the presence of the Vj-gating sensor on the NT has been well documented, the exact conformational changes that Cxs undergo during Vj-dependent gating are unclear. Electron crystallographic studies of Cx26 have provided evidence that the NT domain can also be involved in other types of gating [82]. In these studies, NT domains were proposed to form a “plug”, an occluding particle inside the channel pore. However, because this work was performed on a mutant with defective permeability, and imaged in a buffer with multiple components to favor channel closure (including low pH, carbenoxolone, and high Ca^2+^), it was not possible to say what triggered the observed channel closure [29,82]. Upon generating an N-terminal deletion mutant of residues 2–7, the occluding density in the pore was diminished, suggesting the plug indeed was composed of some or all of the NT domains [98]. Again, as these 2D crystals were generated in almost the same way as the earlier studies, this did not reveal the mechanism by which the NTs would move to occlude the pore. As noted above, experiments using cryoEM and mass spectrometry provide strong evidence for a “ball-and-chain” type of gating in Cx26 GJC, in which six NT domains associate to form an occluding “plug” in the channel pore in response to a decrease in pH [14]. Given the importance of the NT domain, it is conceivable that it could participate in multiple types of Cx GJC and HC gating.

Where the NTs have been resolved in the 6-fold symmetrized cryoEM and crystal structures, they adopt different conformations within the pore itself, highlighting its potential role in channel regulation. In some structures the NTs are unresolved, presumably due to their intrinsic flexibility. Although the conformation of the NT can vary substantially, at least part of the secondary structure is typically helical as assessed by NMR [85,99], X-ray crystallography [30], and cryoEM [33,34,36,37,38,39,40,41], with residues 16–20 forming a short loop connecting NT to M1. The length of this α-helix varies for different Cx structures, from 8 residues for Cx26 to 13 for Cx43. Importantly, the recent deluge of cryoEM structures from seven Cx isoforms (26, 31.3, 32, 36, 43, and 46/50) revealed a diversity of NT conformations (Figure 7). In structures of Cx36 [38], Cx43 [39], and Cx46/50 [41] (Figure 7A), amphipathic NT helices reside inside the channel pore, alongside the channel wall, with their hydrophobic residues forming interactions with residues of M1 and M2 (Figure 7A, bottom panel). In this open conformation, NTs of opposite protomers are approximately 16 Å apart at the narrowest point (Figure 7A, central panels). In the structures of Cx26 [30] and Cx32 [37] (Figure 7B), the NTs adopt an open conformation, but with their ends tilted more towards the inside of the channel, resulting in a slightly narrower Cx pore of ~13 Å (Figure 7B, central panels). This conformation is stabilized by interactions between the NTs of adjacent subunits, as well as interactions of the NT with residues lining the channel wall (Figure 7B, bottom panel).

The available structures of Cx channels in the partially closed state include two structures of Cx43 [39] and a structure of Cx31.3 [36]. One of the partially closed conformations of Cx43 is almost identical to that of Cx31.3 (Figure 7C), with the NTs positioned horizontally at the cytoplasmic pore entrance. An additional, partially closed conformation of Cx43, resolved in a pseudo-HC (a structure of a HC obtained by cryoEM of GJCs), displays ~20° tilting of the NTs towards the inside of the channel pore, as compared to the major, partially closed conformation (Figure 7D). In these partially closed conformations, the NTs are positioned approximately 12 and 10 Å from each other, for the Cx31.3 and Cx43 structures, respectively. The NTs are stabilized mostly by inter- and intra-subunit hydrophobic interactions between NT and M2, inter-subunit interactions between adjacent NTs (Figure 7C,D, bottom panels), and surprisingly, by lipids and lipid-like molecules (Figure 8; described in Section 2.6).

The Trp3 amino acid residue (four in α Cxs) contributes to crucial NT interactions in several structures (Table 2, Figure 7A–C, bottom panels). In the open conformation of the Cx46/50 GJC, the sidechain of Trp3 faces the M1 and M2 helices of the same subunit forming contacts with Ile34, Val79, and Ile82 [41] (Figure 7A, bottom panel). In Cx26, Trp3 (Figure 7B, bottom panel) faces the channel lumen and forms inter-subunit hydrophobic interactions with Met34 in M1 [30]. Other conserved hydrophobic residues, including Leu6(7), Leu9(10), Leu10(11), and Val13(14), stabilize the open conformation of the NT by forming intra-subunit hydrophobic interactions with M1 and M2 residues.

Polar and charged residues of the NT can contribute to inter- and intra-subunit hydrogen bonds and thus further stabilize open conformations. For Cx26, this includes an interaction between Thr5 and Asp2 of an adjacent subunit (Figure 7B, bottom panel), interaction of Gln15 with Thr27 of the same subunit in Cx46/50 (Figure 7A, bottom panel), and a salt bridge between Arg9 and Glu8 of adjacent subunits in Cx36 [38]. The positions of the polar and charged residues are generally not as conserved in the NT, even though they often form important inter- and intra-residue interactions. An example is the unique Gln17 residue in Cx36, which forms an intramolecular hydrogen bond with Gln202 [38]. Variability in the positions of charged residues results in a wide range of overall charges for NTs across different Cxs, which is reflected in their isoelectric points, ranging from 4.4 for Cx50 and 5.5 for Cx36 to 8.4 for Cx26 and 11.7 for Cx31.3 (Table 2). Variability in the positions of charged residues has important consequences for the channel gating mechanism and is likely responsible for granting different Cxs some of their unique properties [8]. Two widely conserved hydrophilic residues are S17(18) and T18(19), which are part of a linker connecting the NT α-helix to M1. This linker varies from 3 to 10 residues and is often stabilized by inter- or intra-subunit hydrogen bonds.

Voltage-driven transitions between fully open and conductance substate(s) are known from physiological studies to involve the NT domain. As noted above, it is possible that the open and partially closed states seen in the cryoEM structures are the structural correlates of fully open and conductance substates seen in single-channel recordings. The substantiation of this speculation awaits further study, but if true, then information regarding the conformational changes in the protein that result in the open ↔ partially closed transitions of the NT domain will be of key importance. In one study of Cx43 [39], open and partially closed structures were resolved, as well as a postulated intermediate structure. On this basis, the authors suggested a transition pathway between the partially closed and open states. In this scenario, the partially closed state (termed “GCN”) contains an α-helical domain in the middle of M1. In the transition to the open state (termed “PCN”), the α-helical segment transitions to a π-helix, accompanied by a significant rotation and bending of the cytoplasmic segment of M1. In addition to altered interactions among TM helices, this change exposes the hydrophobic residues of M1 to the lumen, with which the NT now interacts along the pore wall, forming the open NT configuration. This reaction scheme is inferred from distributions among the two end states (GCN and PCN) and a proposed intermediate state (FIN) seen in the class averages of single particle images. It points to specific, hopefully testable ideas for future investigation.

The effects of the partial pressure of CO_2_ (PCO_2_) on the structure of the Cx26 GJC were recently examined [34]. Three cryoEM maps were obtained at 20, 55, and 90 mmHg PCO_2,_ and molecular models were built. The complete modeling of the NTs in partially closed structures resulted in similar pore diameters of 11 Å. At higher levels of PCO_2,_ the cryoEM densities for the NTs were more complete. The overall architecture of the three structures was otherwise nearly identical. The presence of PCO_2_ appears to stabilize Cx26 GJCs in general, since it was reported (but not shown) that much lower resolution cryoEM maps were obtained in HEPES buffer as compared with sodium carbonate/CO_2_ buffer used in the PCO_2_ comparison studies. When the cryoEM dataset collected on Cx26 GJC at the highest PCO_2_ was subjected to hemichannel-based classification and refinement with C6 symmetry, two HC maps were generated. Two structures were modeled, with one resembling the original structure obtained from the same dataset and one with a unique overall conformation. This new conformation showed the rotation of the cytoplasmic half of a HC with respect to the extracellular half, more defined NT densities connected with a ring of density in the pore, and the movement of M2 away from M1 of the same subunit. These interesting structural changes were not observed for datasets recorded at lower CO_2_ content. Even though the evidence shows that PCO_2_ causes GJC stabilization and induces certain structural changes, the exact mechanism for CO_2_ regulation and the role of the NT in this process remain unclear [34].

### 2.6. Visualization of Lipid-like Densities in Unexpected Places, within the Pore and/or the TMD Helices

For single-particle image analysis, membrane proteins need to be solubilized and stabilized in membrane mimetics such as detergents, amphipols, lipid nanodiscs, or styrene–maleic acid lipid particles (SMALPs). The significant progress made in the past decade in the quality of recorded cryoEM images and data processing algorithms have enabled the determination of membrane proteins structures at near-atomic resolution (~2 Å). Such enhanced techniques applied to Cx channels has enabled the identification of lipid-like densities in the cryoEM maps of Cx26 [34], Cx31.3 [36], Cx32 [37], Cx36 [38], Cx43 [39,40], and Cx46/50 [41]. However, some of the identified densities could not be unambiguously assigned to specific lipids or detergent molecules. Consequently, it was often impossible to differentiate between lipids that were carried over from the cell in which the protein was expressed versus lipids/detergents used to solubilize and stabilize the protein during purification. Typically, these uncertainties arise from the low resolution of the lipid-like densities, possibly caused by their high flexibility, low binding affinity, or a side-effect of particle averaging with imposed symmetry. Nonetheless, in some cases, the fitting of defined lipid or detergent molecules into observed lipid-like densities was possible. Even in the cases where only low-resolution traces of acyl chains could be visualized, new insights into conformational flexibility, gating, and stabilizing lipid interactions could be inferred. Much more remains to be revealed about the interplay between Cx transmembrane and cytoplasmic domains and lipids and how they contribute to Cx channel regulation.

A number of cryoEM maps of the Cx channels have densities outside the channel pore that are ascribed to lipids and/or detergents (Figure 8). In the 1.9 Å cryoEM map of the Cx46/50 GJC in nanodiscs, 15 distinct densities attributed to ordered lipid acyl chains were associated with each subunit [41]. These densities are located at the lateral subunit interfaces, between M4 of one subunit and M2 and M3 of another, within the extracellular leaflet of the lipid bilayer (Figure 8A). Long-range lateral lipid stabilization mediated by Cx46/50 extended ~20 Å beyond the perimeter of the TM domains, corresponding to several layers of boundary lipids. MD simulations suggested the feasibility for Cx-specific lipid stabilization at the extracellular lipid leaflet as well as the hexagonal lipid packing pattern observed in cryoEM. This verified that the lipid stabilization is protein-mediated and not an artifact of the scaffold protein used to generate the nanodiscs. The MD simulations also indicated that Cx46/50 induces a lipid liquid-to-gel transition at the extracellular protein–lipid interface. Reanalysis of the cryoEM images yielded three maps at ~2.5 Å resolution. Each map showed one well-resolved phosphatidylcholine (PC) lipid per subunit (head group and acyl chain) and several lipids for which acyl chains could be visualized. Positions of these three complete PC lipids fitted the hexagonal lipid lattice reported in the original cryoEM map. The authors postulated high-affinity protein binding of the lipid acyl chains and non-specific interactions with the lipid head groups, accounting for difficulties resolving the head groups in the cryoEM maps. Other Cx channel maps visualized multiple acyl chain densities both at the intracellular and extracellular protein–lipid interface, including a 2.4 Å cryoEM map of Cx43 GJC in nanodiscs (Figure 8B) [26], as well as multiple maps of the Cx36 GJC (Figure 8C,D) [38]. A 2.2 Å cryoEM map of the Cx36 GJC also identified phosphatidylglycerol bound to M2 and M3 within the extracellular protein–lipid interface (Figure 8C). Interestingly, a 2.4 Å cryoEM map of the Cx36 GJC displayed an asymmetric distribution of lipid-like densities between hexamers within the cytoplasmic protein–lipid interface (Figure 8D).

Additionally, multiple groups reported lipid densities lining the channel pore, inserting between TM α-helices or interacting with the NT domins. A 2.4 Å cryoEM map of the Cx43 GJC in nanodiscs displayed 12 acyl chains identified in the extracellular half of each HC (Figure 8B). In addition, relatively well-resolved 1-palmitoyl-2-oleoyl-sn-glycero-3-phosphoethanolamine (POPE) and cholesteroyl hemisuccinate (CHS) lipids were found inside the channel pore. Two CHS lipid molecules are bound per subunit in a deep hydrophobic pocket formed between the NT, M1, and M2, and one POPE molecule in each subunit was located between the NTs. The binding of those three lipids masks the solvent-exposed hydrophobic inner surfaces of the channel pore and may stabilize the partially closed conformation of the Cx43 GJC. It is noteworthy that CHS is a soluble cholesterol analog not natively present in the cell membrane, and CHS molecules present in the discussed structures were added during protein purification. In the 2.2 Å cryoEM map of the Cx36 GJC, a pair of CHS molecules and one acyl chain, belonging to either lipid or detergent, were identified in each Cx subunit (Figure 8C) [38]. Interestingly, no interactions of lipids with the NT could be observed as the NTs were not visualized in the map. Another interesting protein–lipid interaction was observed in two Cx36 GJC cryoEM maps (Figure 8D). Two acyl chain densities were bound in a hydrophobic patch formed from the residues of M1 and two adjacent NT α-helices, with Val15 interacting with both of those lipid densities, presumably stabilizing the open channel conformation.

The cryoEM map of the Cx36 GJC also captured an occluded conformation in samples prepared in detergent micelles as well as in nanodiscs [38]. This occluded channel conformation displayed no NT densities, and the channel pore of each HC was blocked by two flat, low-resolution discs of density around the cytoplasmic and extracellular ends of M1 and M2. The occluding densities were also observed in NT-deletion and CL-substitution mutants, largely ruling out the possibility that these cytoplasmic Cx domains contributed to the occluding density. The low resolution of this density has prevented positive identification, which is most likely either the cytoplasmic CT domain or lipid molecules. The authors favored the latter candidate; however, it is unclear how lipids could migrate in and out of the channel pore as the protein transitions between open and occluded states [38]. It also remains a mystery why the Cx36 GJC would have lipids bound inside its pore where other Cx structures obtained from proteins expressed in the same cell types, in the same detergent [39], and/or lipid nanodiscs [35,39] do not display such densities. Nevertheless, this occlusion could indicate that, under some circumstances, lipids may enter connexin channel pores to occlude the permeation pathway. Notably, there are reports that lipids can form conductance-occluding bilayers within other wide- pore channels (CALHM: [100,101] and innexin: [102]). There is also one case in which the movement of acyl chains in and out of the pore lumen is a gating mechanism (TRAAK/KCNK4: [103,104]). 

**Figure 8 biology-13-00298-f008:**
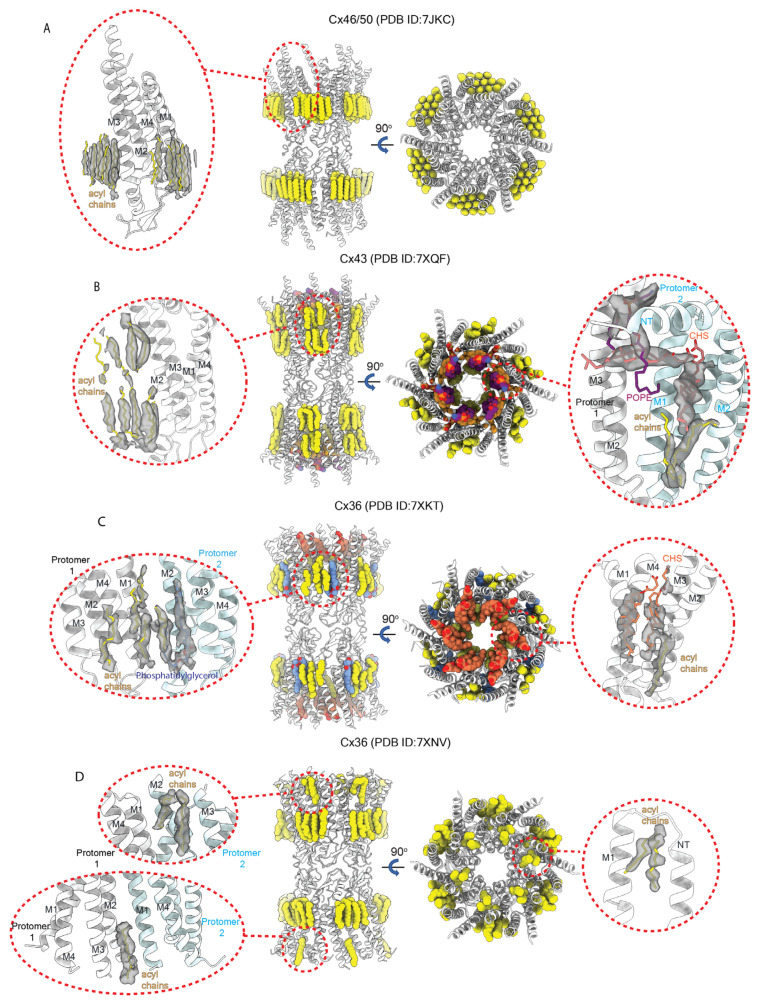
**Interactions between Cx channels and bound lipids as revealed by cryoEM.** Central panels depict a pair of top and side cutaway views for four representative Cx structures ((**A**) Cx46/50 (PDB ID:7JKC), (**B**) Cx43 (PDB ID:7XQF), (**C**) Cx36 (PDB ID:7XKT), and (**D**) Cx36 (PDB ID:7XNV)) with interacting lipids depicted in yellow (acyl chains), orange (CHS), purple (POPE), and navy (PG). Red circles denote details of interactions between lipids and Cx protomers; dark grey indicates lipid density; white and light blue two adjacent Cx protomers.

### 2.7. Functional Implications for Gating and Permeation

The internal walls of eukaryotic wide-pore channels are typically lined by amino acid sidechains rather than backbone atoms. Sidechains may fluctuate among different rotameric conformations and/or be stabilized by internal salt-bridges, or hydrogen bonds, so limiting pore widths based on PDB entries (the deposited structure models) are best considered as relative indicators of the limiting width rather than absolute indicators. In addition, sidechain configuration may be dynamically influenced by the permeants with which they interact, and these interactions can influence whether and how well a molecular permeant can proceed through the open pore (Table 3) [105,106].

Given these considerations, it is perhaps not surprising that the limiting widths of open channels determined from the PDB structures do not vary greatly nor correlate even superficially with the generally accepted size-selectivity values of the various Cx channels for which a moderate to high resolution structure is available [107]. For example, Cx43 is regarded as having the widest pore and Cx32 among the narrowest, but the limiting widths based on the PDB structures do not reflect this (~14 Å vs. ~15 Å, respectively). This suggests that static structures of these channels are not very informative regarding size selectivity, and, conversely, that the basis of size selectivity is likely to be more readily revealed from studies of the dynamics of pore-lining residues (and perhaps backbone atoms) and their interactions with specific molecular permeants. MD studies of Cx26 suggested that molecular permeation was less influenced by highly flexible residues at what seemed to be the “narrowest” part of the pore than by relatively inflexible sidechains elsewhere [105].

Charge selectivity of Cx pores can be considered in two contexts: selectivity amongst current-carrying ions (typically K^+^, Na^+^, Ca^2+^ and Cl^−^) and selectivity amongst cytoplasmic/signaling molecules such as cyclic nucleotides, inositol phosphates, and amino acids, which have moieties with formal and/or partial charges, as well as an overall molecular dipole. In the former category, the primary contributor to electrical coupling is K^+^, due to its much greater concentration in the cytosol compared to the others. The permeability of Ca^2+^ is of course of significant importance in intercellular signaling. In wide pores such as connexins, atomic ions can permeate without significant dehydration (a key basis of selectivity in ion-selective channels) and without obligatory interaction with a narrow selectivity filter with an aperture of the same scale as the ion (c.f., [105]). Because of these considerations, size and charge selectivity among the different atomic ions is much less restrictive than that of ion-selective channels [107].

The situation for molecular permeants differs since they are more likely to come into close proximity with the sidechain moieties lining the pore, and therefore more likely to be influenced by the local electrostatic environment they impart. The nature and degree of these interactions of course depends on the specific permeant, its physical and charge structure, its flexibility, and the specific site(s) of the pore lumen (radial and axial) with which it interacts.

The Cx channel cryo-EM structures alone cannot inform these considerations of molecular permeation, but they can reveal information about the overall electrostatic environment within the pore. Calculations based on the Cx structures in recent studies suggest that, with the exception of Cx36, the electrostatic potential within the pores follows a pattern of positivity at the cytoplasmic entrance and the cytoplasmic side of each hemichannel pore, transitioning to become more negative toward the extracellular/docked portion of the hemichannel. MD calculations based on some of the structures suggest different relative magnitudes of the regions of opposite polarity, with effects on the charge selectivity. For example, the calculations of open Cx46/50 GJCs suggests a slightly higher energy barrier for Cl^−^ relative to K^+^, suggesting a weak cation preference, in line with functional studies [33,108]. Alternatively, the MD studies of partially closed Cx31.3 hemichannels indicates that the cytoplasmic pore entrance and the rest of the pore is positive, with a negative band near the extracellular end of the HC [36]. The MD and experimental studies show that the channel is highly selective for anions, which is attributed fully to the positivity at the cytoplasmic entrance.

The Cx36 pore appears to lack the significant positivity at or near the cytoplasmic pore entrance seen in the other isoforms. In fact, it is somewhat negative near the cytoplasmic pore entrance. However, it retains the region of strong negativity near the extracellular end of each HC [38]. MD shows the channel to be significantly cation selective, which is consistent with functional studies [109].

The data on Cx43 possibly shed some light on changes in charge selectivity as a function of its gating state. One study indicates that in a partially closed state, the positive and negative regions seem to be of similar strength and MD simulations provide little evidence for charge selectivity [40], which is consistent with experimental work [110,111].

Another study proposes that the physiological fully open state of Cx43 consists of channels in which the individual protomers are independently fluctuating between two configurations, one in which the NT is an “open” state against the walls of the pore (the “PLN” state) and another in which the NTs are in a partially closed configuration (the “GCN” state) [39]. It was found that the distribution of the different protomer configurations among the channels was close to that predicted by a random distribution. That is, across the population of channels, there was nearly random mixing of the two protomer types. Calculations and modeling indicated an absence of steric hindrance when the different protomers were mixed within a channel. Further analysis suggested that a channel in which all protomers are in the PLN state has low conductance and high cation selectivity, but that a channel in which half the protomers are in the PLN state and half are in the GCN state has a much higher conductance and is essentially non-selective regarding charge, as observed in physiological studies. These changes in charge selectivity and channel conductance are attributed to the bending and rotation of the NT domain in a GCN-to-PLN transition that both buries basic residues and exposes acidic residues within the pore. From this, it was suggested that in a channel, the individual protomers may each be independently fluctuating among the available states, with all the implied consequences for gating/permeation of the channel [39]. This could contribute to the typically noisy single-channel currents of connexin channels and, in some cases, the occurrence of multiple conductance states. The independence of protomer transitions also fits nicely with physiological studies showing that in a single connexin channel containing a mixture of connexin monomers that respond to opposite voltage polarities, gating transitions to both polarities can be observed, as if each monomer was responding independently to its own preferred voltage polarity [112].

As noted above, a novel finding in some of the structures was the presence of lipid molecules within the permeation pathway, interacting with the protein lining of the pore, and in some cases stabilizing the NT in different conformations. In one instance, for Cx36 [38], lipid-assigned densities were seen to form a plug that completely occluded the pore. In two partially closed structures, densities attributed to acyl chains were found within the pore, interacting with pore-lining helices and thus, contributing to the pore lining. In Cx43 and in Cx36, acyl chains were seen interacting between M1 and M2 and exposed to the pore lumen (Figure 8B, Cx43 7XQF [39]; Figure 8D, Cx36 7XNV [38]). It is presumed that these lipids, like others found embedded among the TM helices and surrounding the protein, contribute to stabilizing the channel structure. Since only acyl chains were resolved, one cannot tell if they correspond to free fatty acids or something more complex (e.g., phospholipids). However, they no doubt contain unresolved charged/polar moieties whose disposition regarding exposure to the pore lumen and possible interactions with pore-lining residues are unknown. Since the positions and interactions of the charged/polar moieties attached to the acyl chains are unknown, one cannot speculate on how they may affect the local charge environment within the pore. The acyl chains interact with hydrophobic regions of the pore walls, so they likely do not alter the electrostatics significantly, but could affect narrowing to some degree and thereby provide a “greasier” wall likely to interact with larger permeants. Since the acyl chains were resolved, one must presume that their interactions with the pore-lining residues are relatively high affinity.

In one Cx36 structure in which the NT was not resolved, a CHS molecule was seen to be exposed to the pore [Figure 8C]; Cx36 7XKT [38]. This cannot be physiological since CHS must have come from the sample preparation, but it does suggest that lipid molecules with compatible properties may interact with the protein and possibly contribute to the pore.

### 2.8. A Summary of Structure/Function Correlations and Insights from High-Resolution Structures

Historically, ion channels were viewed as having two distinct and separable functional properties: *gating*, which determined whether a channel was “open” or “closed”, and *permeation*, which determined the ability of permeants to pass through an “open” channel. For many years, this view had compelling structural and functional support, most dramatically for voltage-sensitive channels, which have voltage-sensing modules linked to but structurally separate from the pathway traversed by ions.

In recent years, the line between gating and permeation has blurred as data emerged showing that gating transitions (defined as changes that could impede or permit permeant flux) could occur within the permeation pathway of an open channel (i.e., at a selectivity filter). This is in addition to previously accepted gating structures near one end of the pore (e.g., the helix crossing of K^+^ channels) that are linked to a separate voltage-sensing module containing an S4 or analogous domain (cf., [113]). The two “gating” sites that physically or electrostatically control whether such a channel is conductive or not must be within the permeation pathway, but their primary dynamics are ultimately driven by linkages, direct or indirect, to domains outside the pore itself. This is not true for Cx channels, nor perhaps for other wide-pore eukaryotic channels.

In connexins, several factors that directly drive gating transitions act within the pore itself. For example, voltage and Ca^2+^ sensing are predominantly mediated by motifs that lie within or are exposed to the pore lumen, as opposed to Ca^2+^- or voltage-sensing modules elsewhere in the protein structure. The sensors for each gating effector appear to be integral components of the “gates” themselves, imposing an obligatory structural linkage between sensing and gating not commonly present in other ion channels, in which the sensors are semi-independent domains distinct from the pore. Because of this, the structural, electrostatic and dynamic features that define connexin channel gating sensitivities also define pore permeability properties, and vice versa [71].

For example, the voltage that drives the HC gating reaction leading to a substate conductance is the voltage sensed *at sites within the pore itself*, not across the protein as a whole (as would be the case for an S4 type of voltage sensing in a voltage-gated ion channel) [114,115,116,117]. The residues at which extracellular Ca^2+^ binds to close hemichannels, and likely junctional channels, lie within the permeation pathway [13,36,72,73]. For this reason, it has been difficult to interrogate separately, at the level of molecular mechanisms, gating and permeability in connexin channels—mutations that affect one property nearly always affect the other.

The structures of Cx channels in which pore and gating elements are resolved at atomic resolution can greatly facilitate the understanding of how this molecular machine operates. The structural studies summarized above, while of different connexins and obtained under a variety of conditions, offer insights regarding some of the functional properties of connexin channels.

As noted, the domains that form the pore lining of all Cx channel isoforms are TM helices M1 and M2. While there is variation in sequence, packing, and limiting pore width across isoforms, overall these structures are similar. Among the various isoforms and conductance states, the charge distribution along the pore axis varies. Differences in the pore width and flexibility of pore-lining sidechains along the pore, and therefore the extent to which molecular permeants interact electrostatically with the protein moieties lining the pore, will also affect charge selectivity among larger permeants.

While the overall luminal structure of the pore is determined by M1 and M2, the NT is a major player in interactions with permeants. In every structure in which the NT has been resolved—including structures that appear to be fully open—the NT is folded into the pore in some fashion. This was inferred from electrophysiological studies as a requirement for the role of the NT in voltage sensing [15,86,87,89]. The NT and its conformations within the pore are key contributors to the physical gating of the channel.

In the open pore cryoEM structures, the NT is flattened against the internal walls, interacting primarily with hydrophobic residues and extending nearly one-half the distance of the hemichannel pore (Figure 7A,B). Thus, the NT affects the electrostatic environment and width of the pore in this region. It also places the N-terminal residue and its charged terminal amine deep inside the pore. The N-terminal amino acid can be acetylated, which removes the positive charge of the N-terminal amine [14,78]. If the N-terminal residue is not acetylated, its charge profoundly alters the charge selectivity of the pore [31,33]. In the structures discussed, an intra-pore NT was resolved in the fully open channels of Cx26 [14,30], Cx32 [37], Cx36 [38], Cx43 [39], and Cx46/50 [41]. However, acetylation status could not always be assessed. In those cases in which the N-terminal domain is unresolved, or in which the acetylation status of the N-terminal amine within the pore is unknown, assumptions and modeling of the size and charge selectivity of the pore that omit these elements are potentially incorrect.

One striking feature emerging from the structural analysis that we present is the distinction between “open” and “partially closed” channels, based on the limiting pore diameters determined from the PDB entries, which themselves are functions of the position of the NTs (Table 2). As noted, this review considers open channels as those having limiting pore widths ~13 Å and greater, with partially closed channels as those having limiting pore widths in the range ~ 9–12 Å. In some cases, this delineation is aided by two structures of the same channel with the NT in different configurations, one with a wider limiting diameter than the other (Table 3).

In the partially closed states, the NTs were found in two configurations within the pore, both of which restrict the permeation path. In one configuration, the NT is separated from the hydrophobic patch on the pore wall that it occupies in the open state and adopts an orientation parallel to the plane of the membrane, with the N-terminal end pointing toward an adjacent subunit [37]. In the second configuration, it is also oriented parallel to the plane of the membrane but pointing directly toward the center of the pore [36,38,39,40] (Figure 7C,D). As described above, each of these partially closed conformations are stabilized by interactions with residues of M1 and in some cases lipids. In some studies, open and partially closed NT configurations are seen in the same sample preparation, suggesting an equilibrium between these two NT states. Put another way, we may be seeing the open and partially closed configurations of the NT as two end-states of a dynamic equilibrium. The conformational and energetic pathway that links them remains cryptic.

A ubiquitous feature of Cx channel gating is transitions between a maximal (i.e., fully open) conductance state and one or more conductance substates (i.e., states with a significantly smaller electrical conductance). Gating to substates is often referred to as “Vj-gating” and based on extensive data is attributed to voltage-driven movement of the NT [86,118]. It is intriguing to attribute the partially closed states and the corresponding configurations of the NTs to be a structural basis of the subconductance states, especially since they all feature NT domains in configurations that reduce the limiting pore diameter. At present, this is just an interesting hypothesis (explicitly suggested in [39]) and awaits further study for substantiation.

Note that the NT was resolved in all of the partially closed structures; no narrowed pores were seen in the absence of resolved NTs. Since the pore is not occluded in the partially closed states, and the NT is exposed to the pore lumen in different configurations from that in the open state, the permeability properties of the pore of the presumed substate may differ from those of the open state in ways other than size restriction. Indeed, there is evidence that Vj-gated substates, in addition to the restricted permeation of large permeants that is expected for a narrowed pore, also can have altered charge selectivity [111,119,120].

As mentioned previously, the other form of Cx channel gating closes the pore fully. This has been most dramatically studied in undocked HCs and has been shown to be tightly regulated by extracellular Ca^2+^ and voltage [118,121,122]. While the sensitivity to these factors is well characterized, and in the case of Ca^2+^, the site of Ca^2+^ action located [72,73], the structural mechanisms remain elusive (but see below). This type of gating is called “loop-gating” or “slow-gating” because it was thought to be mediated by the extracellular loops and because the transitions into and out of the closed state are slow [118,123]. Unfortunately, cryoEM studies to date do not shed light on the loop-gating transitions or mechanisms. In some cases, it is noted that the extracellular loops show some flexibility, but no more than that. Investigation of the structural basis of this important gating transition (which keeps undocked hemichannels closed under most normal conditions) awaits further study.

Discussion of the Cx pore properties must consider whether the NT resides within the pore. The NT is not resolved in some of the cryoEM structures, but appears to be resolved in several others, and in all those cases it is within the pore. What are we to make of the instances in which it is unresolved in cryoEM structures? If it is in the pore but unresolved, it could be highly mobile within the pore, rapidly fluctuating amongst many configurations, without occupying of any of them sufficiently to be resolved. However, it is clear from the structures in which the NT *is* resolved that it can be stabilized in the pore in at least three resolvable configurations by interactions with pore-lining sidechains and/or lipids. Given this, it seems unlikely that if it was in the pore it would not favor *any* of the stable conformations sufficiently to be resolved if the required resolutions were attained. Another option is that the NT is not folded into the pore but is extended from the cytoplasmic end of the pore into free solution outside the plane of the membrane with unconstrained configurations. However, it is clear from physiological studies that the NT, particularly residues toward the N-terminus, can sense and respond to membrane voltage. If it is outside the plane of the membrane, it cannot sense the membrane or pore voltage field and therefore, cannot respond and relocate in response to it to mediate its well-documented role in Vj-gating. Previous computational studies of permeation that reproduced single-channel current–voltage relations, and specific aspects of charge and molecular selectivity of open connexin channels, explicitly included the NT in the pore, and indicated the involvement of the NT in those processes (for example, [31,105,106]). We therefore favor the view that the NT resides within the pore in a normally functioning connexin channel.

## 3. Conclusions and Future Prospects

After about 15 years of effort, cryoEM of 2D crystals of recombinant GJCs yielded 3D maps at 7.5 and 5.7 Å resolution, in 1999 and 2004, respectively [27,28]. The GJC had an elegantly simple architecture with 24 helices within each hemichannel, end-to-end docking to form the GJC, and a continuous wall of protein precluding communication with the extracellular space [27,28]. The maps revealed the tilt and packing of the TM α-helices in the four-helix bundle design of each subunit. An additional decade was required to generate 3D crystals that enabled the determination of X-ray structures at ~3.5 Å resolution [13,30]. The electron density maps finally revealed the side-chain interactions that explained the stability at the docking interface between hexamers [13,30], and the coordination of 12 Ca^2+^ ions between adjacent subunits conferring dramatic electropositivity that precluded permeation by cations under states of tissue injury associated with Ca^2+^ overload [13,30]. In spite of these breakthroughs, it was clear that the generation of 3D crystals of other Cx isoforms and disease-causing mutants was not at all straightforward.

A major breakthrough for cryoEM was the invention of direct electron detectors with exquisite sensitivity that allowed the use of exceedingly dim beams to minimize radiation damage. Combined with improvements in the optics provided by the 300 kV Titan Krios electron microscope and more robust image processing software enabled the determination of high-resolution Coulomb potential maps by single-particle image analysis that rivaled electron density maps derived by X-ray crystallography [124,125,126,127]. This set the stage for the determination of more than 50 high-resolution cryoEM maps of seven Cx isoforms, spanning four Cx families, under a variety of experimental conditions. These remarkable achievements have involved at least six distinct research groups on three continents.

A special advantage of cryoEM is that macromolecular complexes can be examined in a near-native state by freeze trapping in vitrified physiological buffer. In the future, cryoEM will continue to play a critical role in structural biology, which will certainly be augmented by the use of AI-based structure predictions and MD simulations. CryoEM currently provides a limited number of high-resolution molecular “snapshots”. It is also noteworthy that the structures determined by cryoEM and X-ray crystallography enforce 6-fold symmetry, which may be underestimating Cx conformational heterogeneity. Analysis of massive data sets by time-resolved cryoEM and single-particle analysis without enforcing symmetry may reveal the full range of channel conformational states [128]. Indeed, MD simulations show deviations from symmetry within 50–100 ns [13,30]. In addition, it is quite possible that within an HC the NTs may have multiple conformations. This is exemplified by the analysis of pannexin 1 channels in which 3–4 CTs have the capacity to block the pore, which would deviate from the overall symmetry of the channel [129]. Unfortunately, the individual subunit structures in heteromeric Cx46/Cx50 GJCs could not be well resolved, even though the map resolution was 1.9–2.5 Å. This implies that the analysis of heteromeric Cx channels will require the generation of cleavable concatemers, as we did for pannexin 1 channels with variable numbers of C-tails [129].

Instead of solubilizing membranes with detergents to isolate GJCs and HCs for reconstitution into membrane mimetics, it is now possible to isolate integral membrane proteins with their native lipids by using polymers that remove patches of membrane, exemplified by analysis of the heterodimeric integrin αIIbβ3 [130]. Alternatively, it is also possible to solve structures of integral membrane proteins within small vesicles without removal from the native membrane [131]. These two approaches may enable the determination of near-native structures, as well as Cx channel structures with interacting proteins such as Z01 [132]. Advancements in cryotomography may enable in situ structural biology providing insight into the assembly and turnover of gap junction plaques [133].

Notably, no Cx structure has been reported in which the M2–M3 cytoplasmic loop (CL) or cytoplasmic tail (CT) have been visualized by X-ray crystallography or cryoEM, presumably due to conformational flexibility and/or the lack of secondary structure. In contrast, all Cx structures include modeled conformations for EL1 and EL2. However, beyond HC docking to form junctional channels, there are few significant differences between ELs across the structures. Of course, this structural conservation may arise from docking within GJCs, in which the ELs form a tight docking interface, whereas the Els in hemichannels may have more conformational flexibility [35,45]. A molecular basis for loop (or slow) gating in Cx channels will likely only arise from a comparison of undocked HC structures from across isoforms and in the presence and absence of loop gating regulators, such as Ca^2+^ and voltage. Lastly, as described in detail above, the configuration of the NTs could be modeled in a range of conformations. For example, from cryoEM structures of Cx43 a two-state configuration could be modeled, with the NT either parallel to and lining the pore wall (the “PLN” conformation) or lying parallel to the plane of the membrane and pointing at the center of the pore axis (the “GCN” conformation). Though awaiting experimental assessment, these could be the structural configurations directly associated with subconductance states observed from the electrophysiological recordings of Cx channels. However, we lack any structural information for the NT in some of the deposited models. As its role in Vj-gating is required, it is paramount that all of its lowest energy conformational states are elucidated across as many isoforms as possible.

To gain insight into the conformational dynamics of GJCs and HCs, future studies could employ techniques such as structure prediction and molecular dynamics simulation (MD) [48], crosslinking mass spectrometry (XL-MS) [134], and hydrogen/deuterium exchange mass spectrometry (HDX-MS) [135]. For instance, our use of molecular dynamics and electrostatic calculations suggested the novel mechanism of Ca^2+^-mediated electrostatic gating of Cx26 [13,30]. In contrast, pH gating is mediated by a steric “ball-and-chain” mechanism [14]. To our knowledge, the analysis of the physiologic and low pH conformations of Cx26 was the first application of XL-MS and HDX-MS in the Cx channel field, and the results reinforced our model for the ball-and-chain mechanism suggested by our cryoEM maps at 7.5 Å resolution. In addition, DEER electron paramagnetic resonance (EPR) spectroscopy [136,137] and fluorescence resonance energy transfer [138,139] are particularly useful for proximity measurements in the 5–10 Å and 15–50 Å range, respectively. Fortunately, the Sali lab has developed approaches for integrating data derived from multiple methods [140,141].

There are 14 isoforms for which a high-resolution structure has yet to be resolved. As Cx channels are required for a range of physiological processes across a broad panel of tissues, further structures are needed, especially to address Cx-associated pathologies, which often involve a disease-causing mutant. To facilitate drug discovery and design, structural studies will need to include disease-causing Cx mutants, for which there are limited data [36,37]. The future is both daunting and exciting.

In a review article now a decade old, we stated “Recent technical advancements in EM, X-ray crystallography and computational simulation create unprecedented synergies for integrative structural biology to reveal new insights into heretofore intractable biological systems” [142]. Combined with a full armamentarium of approaches to examine conformational dynamics, the stage is set for the continued exploration of the structure, dynamics, and regulation of connexin channels in health and disease. To our knowledge, no Cx channel structures have been determined with bound small molecules, peptides, or antibodies that modulate channel activity. Indeed, we anticipate that drug discovery of connexin channels will be an important theme in the next 15 years [143,144].

## Figures and Tables

**Figure 1 biology-13-00298-f001:**
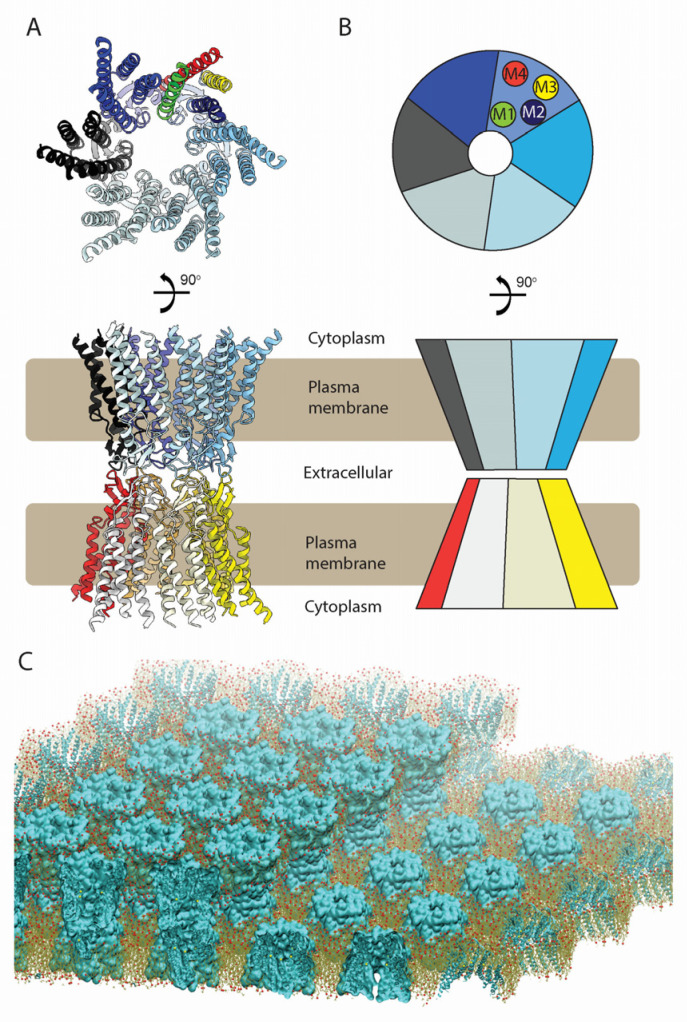
**Gap junction structure, nomenclature, and plaque assembly in the membrane.** (**A**) Top and side view of the Cx26 gap junction channel (PDB ID:5ER7). The hemichannel is comprised of 6 protomers, and hemichannels dock end-to-end between apposed cells to form the intercellular wide-pore channel. (**B**) Schematic representation of a gap junction channel. M1–M4 α-helices are colored green (M1), navy (M2), yellow (M3), and red (M4) in the upper panels of (**A**,**B**). Individual protomers are colored in different shades of blue for one hexamer, while the protomers of the hexamer from the adjacent cell are colored from yellow to red. (**C**) Representation of a gap junction plaque comprising gap junction channels that pack with quasi-hexagonal symmetry.The Cx gene family is diverse, with 21 identified members in the sequenced human genome, and 20 in the mouse (19 of which are orthologs with human Cx). Connexins are commonly named according to their molecular weights (e.g., Cx26 for the 26 kDa isoform), and their molecular masses range between 25 and 60 kDa. Alternatively, Cxs are also classified into five families based on their sequence homology—α, β, γ, δ, and ε, followed by an identifying number (e.g., GJA1 refers to Cx43) [5].

**Figure 2 biology-13-00298-f002:**
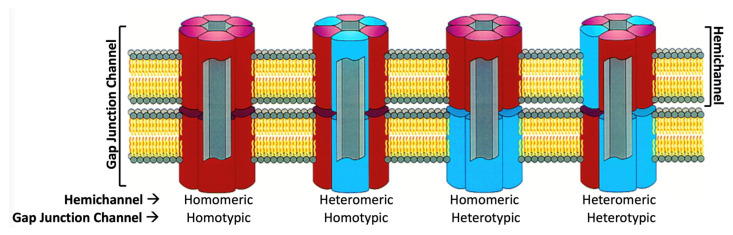
**Oligomeric states of gap junction channels and hemichannels assembled from two isoforms (red and blue).** HCs can exist as homomeric or heteromeric hexamers, whereas GJCs can exist as homotypic or heterotypic dodecamers. (Adapted from [9]).

**Figure 3 biology-13-00298-f003:**
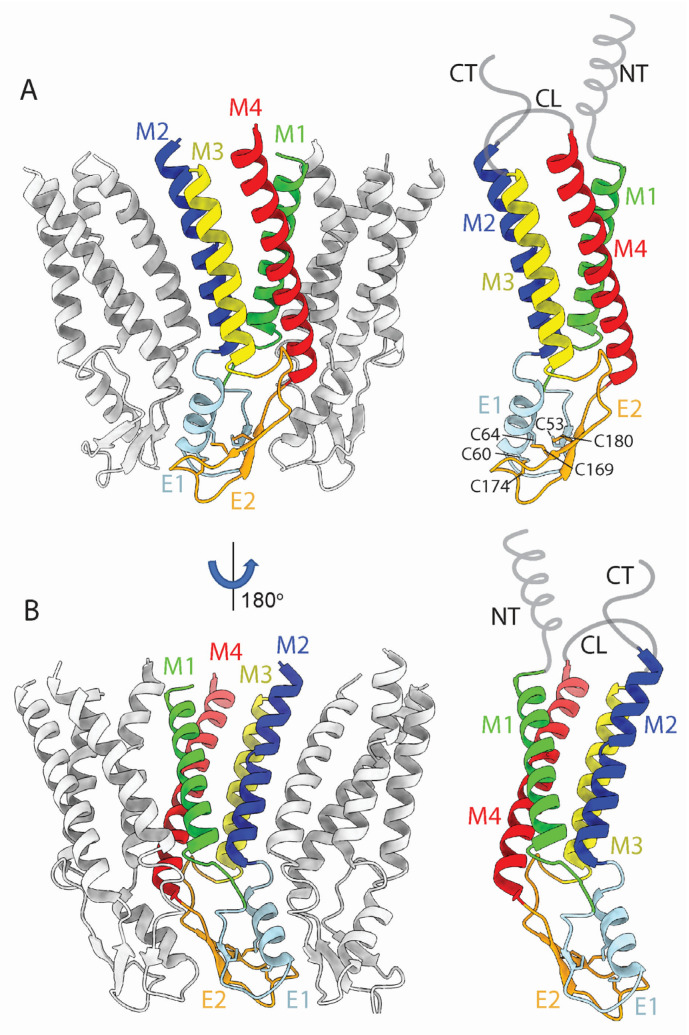
**Cx protomer structure in the context of the Cx26 hemichannel (PDB ID: 5ER7 [13]).** (**A**) depicts a view within the lipid bilayer showing the perimeter M3 and M4 α-helices of the protomer. (**B**) depicts a view from the center of the hemichannel showing the pore-lining M1 and M2 α-helices of the protomer. All connexins are predicted to have a topological structure similar to that of Cx26, with four transmembrane domains (M1–M4), two extracellular loops (E1 and E2), a cytoplasmic loop (CL), an amino terminal (NT), and a carboxyl terminal (CT) domain. The positions of disulfide bonds between E1 and E2 are indicated in the right panel of (**A**). Grey lines represent structurally unresolved CL and CT domains. A region of NT may fold as an α-helix that can have multiple positions.

**Figure 4 biology-13-00298-f004:**
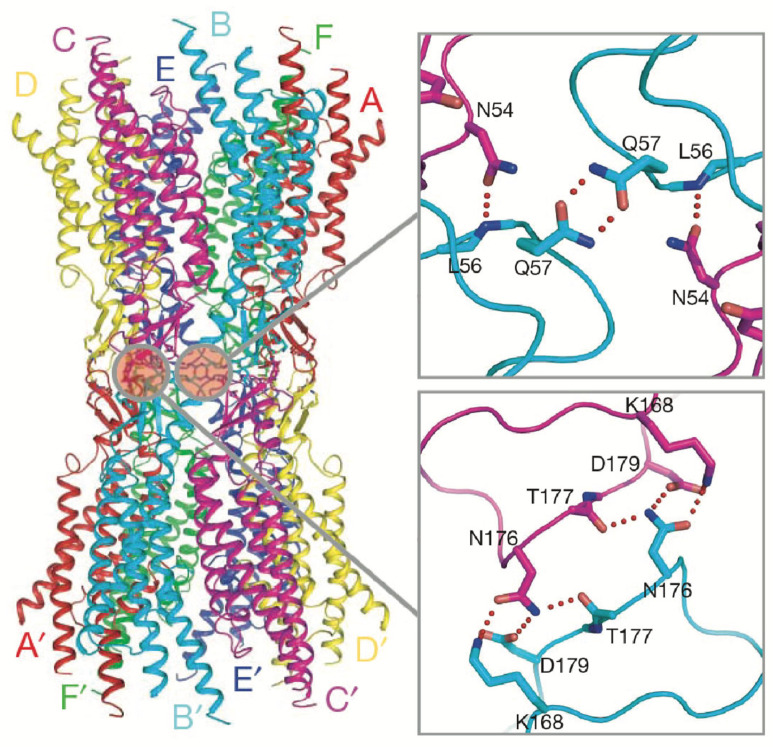
**Docking between hemichannels is the raison d’être of gap junction channels.** The subunits in the side view are labelled A to F and A′ to F′, each in the same color. The enlarged views show the hydrogen bonding interactions in E1 (**top**) and E2 (**bottom**) that stabilize the dodecameric gap junction channel (Adapted from [30]).

**Figure 6 biology-13-00298-f006:**
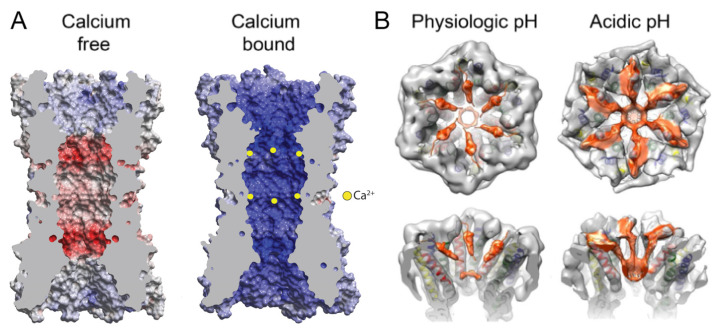
**Gating mechanisms of gap junction channels during tissue injury, accompanied by Ca^2+^ overload and acidic pH.** (**A**) Binding of 12 Ca^2+^ ions results in a channel pore that is highly electropositive, resulting in the electrostatic block of cation permeation such as K^+^ [13]. Electrostatic potential surfaces with positive and negative electrostatic potentials are shown in blue and red, respectively (color scale is −15 to +15 kTe^−1^). The protein interior is grey. Yellow spheres indicate that Ca^2+^ ions bind between adjacent subunits in each hemichannel. Hemispheric binding at the TMD/ECD interface directly exposes the Ca^2+^ ions to the aqueous pore, thereby maximizing their electrostatic positivity. (**B**) Similarly, acidic pH results in steric block of the channel pore by a “ball-and-chain” mechanism, in which the ball is composed of the NT domains. (Adapted from [13,14]).

**Figure 7 biology-13-00298-f007:**
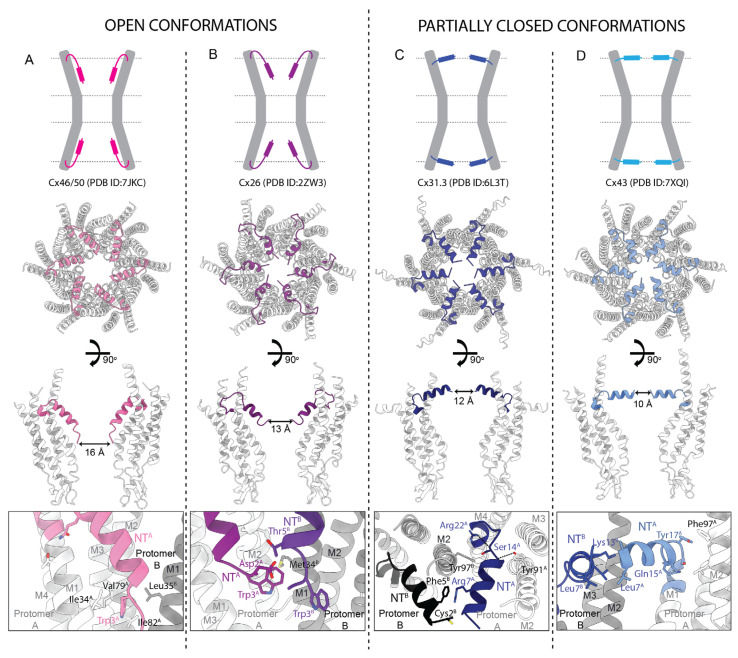
**Distinct conformations of the NT domains resolved in the cryoEM structures of Cx channels.** Upper panels—schematic representation of open (**A**,**B**) and partially closed (**C**,**D**) NT conformations. Central panels—a pair of top and side cutaway views for four corresponding Cx structures with differing NT conformations ((**A**) Cx46/50 (PDB ID:7JKC), (**B**) Cx26 (PDB ID:2ZW3), (**C**) Cx31.3 (PDB ID:6L3T), (**D**) Cx43 (PDB ID:7XQI)); NTs shown in color, M1–M4 and E1–E2 shown in grey; narrowest pore diameter indicated. Bottom panels—selected details of NT interactions. References: Cx26 [30], Cx31.3 [36], Cx36 [38], Cx43 [39] and Cx46 [41].

**Table 1 biology-13-00298-t001:** **Summary of high-resolution Cx structures published from 2009 to 2023.** HC: structures of hemichannels obtained from cryoEM of purified hemichannels; pseudo-HC: structures of hemichannels obtained from cryoEM of purified junctional channels; GJC: structures of junctional channels obtained from purified junctional channels; asymmetric GJC: structures of junctional channels in which the component hemichannels are in different structural states. Solvents: UDM: n-Undecyl-β-D-Maltopyranoside; FA3: Façade-EM; LMNG: Lauryl Maltose Neopentyl Glycol; CHS: Cholesteryl Hemisuccinate; GDN: Glyco-Diosgenin. Lipids: POPE: Phosphatidylethanolamine; POPC: Phosphatidylcholine; DMPC: 1,2-Dimyristoyl-sn-Glycero-3-Phosphocholine. +BRIL: CL replaced by cytochrome b562RIL residues 21–128.

Cx	Gene	Structures	Technique	Res. [A]	Symmetry	Solvent	Conformation	Features	Gating	Mutations	Refs.
26	GJB2	GJC	X-ray	3.5	C2, C6 NCS	Detergent: UDM	Open	NT			[30]
26	GJB2	GJC	X-ray	3.3–3.8	C2, C3	Detergent: FA3	Open, Ca^2+^-bound		Ca^2+^		[13]
26	GJB2	GJC, pseudo-HC	CryoEM	1.9–2.2	C6, D6	Detergent: DDM	Partially closed	NT, lipids or detergents, waters	PCO_2_		[34]
26	GJB2	GJC	CryoEM	4–7.5	D6	Amphipol: A8–35	Open, Closed	NT	pH		[14]
26	GJB2	HC	CryoEM	4.2	C6	Nanodisc: Soy Lipids	Open			N176Y	[35]
31.3	GJC3	HC	CryoEM	2.3–2.6	C6	Detergent: LMNG	Partially closed	NT, lipids or detergent, waters	Ca^2+^	R15G	[36]
32	GJB1	GJC, HC	CryoEM	2.1–3.7	D6, C6	Detergent: Digitonin	Open, Partially closed	NT, lipids or detergent, waters		W3S, R22G	[37]
36	GJD2	GJC, asymmetric GJC	CryoEM	2.2–7.2	D6, C6, C1	Detergent: LMNG/CHS Nanodisc: Soy Lipids	Open, Lipid-occluded	NT, lipids or detergent, waters		+BRIL Δ1–8 +BRILΔ1–16	[38]
43	GJA1	GJC, asymmetric GJC, pseudo-HC	CryoEM	2.4–4	D6, C6, C1	Detergent: LMNG/CHS, GDN Nanodisc: Soy Lipids, POPE/CHS	Open, Partially closed	NT, lipids or detergent, waters		Δ257–382	[39]
43	GJA1	GJC, HC	CryoEM	2.3–4	D6, C6	Detergent: Digitonin Nanodisc: POPC	Partially closed	NT, lipids or detergent,			[40]
46/50	GJA3, GJA8	GJC	CryoEM	3.4–3.5	D6	Amphipol: A8–35	Open	NT			[33]
46/50	GJA3, GJA8	GJC	CryoEM	1.9–2.5	D6	Nanodisc: DMPC	Open	NT, lipids or detergent, waters			[41]

**Table 2 biology-13-00298-t002:** Comparison of the sequences of the N-terminal domains (NTs) of WT human Cxs for which the structures of the NTs have been resolved (modified from [83]). The Cxs are grouped according to sub-families (GJA, GJB, etc.). The conformation of the NT in each structure is listed as either open (O) or partially closed (PC). The pI denotes predicted isoelectric point of the NT. Gaps were allowed to maximize the identities among sequences. Numbers across the top indicate the amino acid number for the β-Cxs (GJBs) and for the α-Cxs (GJAs). Polar amino acids are green, hydrophobic amino acids are orange, negatively charged amino acids are red and positively charged amino acids are blue. Grey indicates residues not resolved in the structure.

AA Number	(β)	NT Conformations	pI	1		2	3	4	5	6	7	8	9	10	11	12		13	14	15	16	17	18	19	20	21	22
AA Number	(α)			1	2	3	4	5	6	7	8	9	10	11	12	13		14	15	16	17	18	19	20	21	22	23
Cx26	GJB2	O, PC	8.4	M	-	D	W	G	T	L	Q	T	I	L	G	G	-	V	N	K	H	S	T	S	I	G	K
Cx31.3	GJC3	PC	11.7	M	-	C	G	R	F	L	R	R	L	L	A	E	-	E	S	R	R	S	T	P	V	G	R
Cx32	GJB1	O, PC	10.8	M	-	N	W	T	G	L	Y	T	L	L	S	G	-	V	N	R	H	S	T	A	I	G	R
Cx36	GJD2	O	5.5	M	G	E	W	T	I	L	E	R	L	L	E	A	A	V	Q	Q	H	S	T	M	I	G	R
Cx43	GJA1	O, PC	8.2	M	G	D	W	S	A	L	G	K	L	L	D	K	-	V	Q	A	Y	S	T	A	G	G	K
Cx46	GJA3	O	5.4	M	G	D	W	S	F	L	G	R	L	L	E	N	-	A	Q	E	H	S	T	V	I	G	K
Cx50	GJA8	O	4.4	M	G	D	W	S	F	L	G	N	I	L	E	E	-	V	N	E	H	S	T	V	I	G	R

**Table 3 biology-13-00298-t003:** **Comparison of pore diameter in Cx GJCs and HCs, as measured from their deposited structures**. The limiting (narrowest) width of each isoform was determined from PDB entries as that between the identical atoms of opposite subunits in the narrowest section of the pore. Diameters are reported in Å, rounded to whole numbers.

Open Conformation			
Cx Isoform	Pore Diameter (Å)	GJ/HC	References
Cx26	15	GJ	[30]
Cx32	15	HC	[37]
Cx36	15	GJ	[38]
Cx43	14	GJ	[39]
Cx46	16	GJ	[41]
Cx50	15	GJ	[41]
**Partially closed conformation**			
Cx31.3	12	HC	[36]
Cx32 W3S	10	HC	[37]
Cx32 R22G	10	HC	[37]
Cx36	11	GJ	[38]
Cx43	10	GJ	[39]
Cx43	10	GJ	[39]
Cx43	10	GJ	[40]
Cx46	10	GJ	[41]
Cx50	9	HC	[41]

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
