# Peer review of "Connexin Gap Junction Channels and Hemichannels: Insights from High-Resolution Structures"

_biology, 2024, doi:10.3390/biology13050298_

Round 1

Reviewer 1 Report

Comments and Suggestions for Authors

Cell communication is essential for various biological processes, including development, physiology, and responses to disease and injury. Connexins (Cxs) play a crucial role in facilitating this communication, acting as both hemichannels (HCs) and gap junction channels (GJCs) to allow the passage of molecules between cells and their environment. Dysfunction in GJC and HC communication can contribute to various pathological conditions. Recent advances in imaging techniques have revealed structural insights into Cx isoforms, highlighting their role in forming tight seals between cells. The authors have extensively reviewed the literature in the manuscript about the structures available and the general architecture of connexins. The manuscript is well written.

Minor comments:

In Figure 1A, marking the helices M1-4 like in 1B would be helpful. It would help the readers to orient better.

Please elaborate a little on the nomenclature of the Cx isoforms.

In Figure 2B, does the RMSD imply that M1 is the most flexible part of the structure and M2 is the least flexible? Please elaborate on the interpretation of the graph.

From line 142 onwards, the text becomes rich in information. Is it possible to include a schematic of how the plaque is formed?

In terms of target for drug discovery, which is the most critical region to target? Have any efforts been made in this direction?

Reviewer 2 Report

Comments and Suggestions for Authors

This review manuscript provides a brief background introduction to the connexin gap junction channels and hemichannels and summarizes in detail the major progress in their structural studies during the past 15 years. It has been well-thought-out and carefully organized and covers the overall structural architecture, sequence differences, mechanisms for gatings, multi-confirmations, density in the liquid region, functional insight, and perspective. Besides comparing the structural details, it is particularly insightful that the review discusses the models of functional mechanisms proposed by each structural work. The figures of the structures are well prepared. The schematic Figure 4 is particularly informative. I recommended the review manuscript be published after minor editing.

Comments on the Quality of English Language

Below are some example errors that require authors’ attention.

Page 12, Line 458: “Fig.4B, central panels)..” should be “Fig.4B, central panels).”, i.e. the extra dot should be removed.

Page 19, Line 746: The foot of the subtitle “(8) A summary of structure/function correlations and insights from high-resolution” should be in bold and stay at the beginning of the following paragraph consistent with other subtitles.

Page 22, Line 920: “……cleavable contcatamers,” should be “……cleavable concatemers,”

Page 22, Lin 937: “in which the ELs form a tight docking inerface,” should be “in which the ELs form a tight docking interface,”, i.e. error in spelling.
